# Metabolomics Analysis Reveals the Effect of Two Alpine Foliar Diseases on the Non-Volatile and Volatile Metabolites of Tea

**DOI:** 10.3390/foods12081568

**Published:** 2023-04-07

**Authors:** Yuhe Wan, Yuxin Han, Xinyi Deng, Yingjuan Chen

**Affiliations:** Department of Tea Science, College of Food Science, Southwest University, Chongqing 400715, China

**Keywords:** tea diseases, metabolomics analysis, non-volatiles, volatiles

## Abstract

Blister blight and small leaf spots are important alpine diseases that mainly attack tender tea leaves, affecting tea quality. However, there is limited information on the effect of these diseases on tea’s non-volatile and volatile metabolites. Metabolomic analysis based on UHPLC-Q-TOF/MS, HPLC and GC/MS was used to reveal the characteristic chemical profiles of tea leaves infected with blister blight (BB) and small leaf spots (SS). Flavonoids and monolignols were non-volatile metabolites that were enriched and significantly changed. Six main monolignols involved in phenylpropanoid biosynthesis were significantly induced in infected tea leaves. The accumulation of catechins, (−)-epigallocatechin gallate, (−)-epicatechin gallate, caffeine, amino acids and theanine were significantly decreased in both diseased tea leaves, while soluble sugar, (−)-epigallocatechin and phenol-ammonia were obviously increased. Among them, the amounts of sweet and umami-related soluble sugar, sucrose, amino acids and theanine were much higher in BB, while bitter and astringent taste-related catechins and derivatives were much higher in SS. Volatiles analysis showed that volatiles content in SS and BB was significantly decreased, and styrene was significantly induced in blister blight-infected tea leaves. The results indicate that the type and amount of volatiles were highly and differentially influenced by infection with the two alpine diseases.

## 1. Introduction

Tea, as a healthy beverage, is rich in various secondary metabolites, including polyphenols, amino acids, alkaloids, polysaccharides, lipids and volatiles, which are widely consumed for their unique flavor and health-promoting benefits [1,2,3]. Numerous studies have validated that biotic and abiotic stresses, including pathogen infection (such as *Colletotrichum*, *Exobasidium*), insect attack (tea geometrid, tea green leafhoppers), mechanical damage and environmental conditions can significantly affect the quality of tea by changing the content and composition of metabolites, especially the volatile compounds in plants [4,5,6,7]. Previous studies reported that tea leaves infested with tea green leafhoppers (*Empoasca onukii* Matsuda) endow oolong tea with characteristic volatile compounds that contribute to the unique aroma of oolong tea [5]. Shading is a common cultivation measure on tea plants, which can obviously improve tea quality by increasing the amount of chlorophyll, amino acids and phenylpropanoids/benzenoids [4,8]. Tea plants have abundant caffeine and phenolic compounds, which act as a vital role in disease resistance, and the amount of these compounds in tea plants is usually increased in response to pathogen infection [9,10].

Tender tea buds and leaves are the raw materials for tea processing. Blister blight—caused by a fungus *Exobasidium vexans*—and small leaf spot disease showing pinhead-like spots—caused by a fungus *Didymella segeticola* var. *Camellia*—are considered to be the most damaging diseases worldwide. They mainly attack young organs and tissues, especially tender tea leaves, which seriously affects tea quality [11,12]. So far, blister blight disease has been reported to occur in almost all tea plantation regions of Asia, and small leaf spot disease is a newly reported disease in southwest China [11,12,13,14,15]. Previous studies reported that tea made from blister blight-infected tea leaves is fragile and has an obvious bitter taste, and the content of tea polyphenols and catechins is decreased significantly [16]. In China, both blister blight and small leaf spot diseases infect tender shoots, petioles and young stems of tea bushes, which occur widely throughout the alpine tea gardens in southwest China, including Guizhou, Sichuan province and Chongqing city in recent years [12,14,15]. However, the effect of both diseases on tea quality has not been systematically studied. In our previous studies, we found that tea quality was negatively affected by disease infection, in which tea leaves infected with small leaf spots had a strong bitter taste, while tea leaves infected with blister blight had an enhanced sweetness with a light fishy taste (data not published). To reveal and compare the effect of blister blight and small leaf spot diseases on the non-volatile and volatile metabolites of tea, the differential metabolites between healthy and diseased tea leaves were comprehensively investigated. Ultra-high performance liquid chromatography-quadrupole-time of flight mass spectrometry (UHPLC-Q-TOF/MS), high-performance liquid chromatography (HPLC) and gas chromatography tandem mass spectrometry (GC-MS) were applied for nontargeted and targeted metabolomic analysis.

## 2. Materials and Methods

### 2.1. Sample Preparation

From 2018 to 2020, pinhead-like leaf spots caused by *D. segeticola* and blister blight disease caused by *E. vexans* were simultaneously observed in the commercial tea plantations located in Quxian, Sichuan province (30°54′ N, 107°09′ E) of China, lying at an elevation of 1100–1200 m above sea level. The tea garden was mainly planted with Sichuan middle- and small-leaf tea tree population varieties, covering an area of 1000 mu. In April 2020, tea fields infected with pinhead-like leaf spots and blister blight were used for the present study. Tea fields infected with *D. segeticola* are about 200 m apart from the fields infected with *E. vexans*, and a large number of pine trees are densely planted around each tea field. Fresh tea shoots (one bud and two leaves) infected with blister blight and small leaf spots were harvested to investigate the effect of two diseases on the metabolites of tea, while healthy tea shoots were used as a control. A total of 200 g of each tea sample was collected from more than forty tea plants, and the sampling process was repeated three times. Collected tea samples were fixed in a tea dryer for about 1 h, and finally were milled and stored at −80 °C for metabolite analysis.

### 2.2. Non-Volatile Metabolite Analysis

Each tea sample (50 mg) was accurately weighed and extracted with a 400 µL methanol:water (4:1, *v/v*) solution. The mixture was treated with a high-throughput tissue crusher Wonbio-96c (Shanghai wanbo biotechnology Co., Ltd., Shanghai, China) at 50 Hz for 6 min, then vortexed for 30 s, treated with ultrasound (40 kHz) for 30 min at 5 °C, placed at −20 °C for 30 min, then centrifuged at 13,000× *g* at 4 °C for 15 min. The supernatant was analyzed for metabolites using ultra performance liquid chromatography coupled with time-of-fight mass spectrometry (UPLC-TOF/MS) system. The non-targeted metabolomic analysis was performed to study the differential metabolite compositions between healthy tea leaves (HT) and diseased tea leaves. Diseased tea samples included tea leaves infected with small leaf spot disease (SS) and tea leaves infected with blister blight disease (BB). A quality control (QC) sample was used for a part of the system adjustment; it was prepared by mixing equal volumes of HT, BB and SS samples. After UPLC-TOF/MS analyses, the raw data were imported into the Progenesis QI 2.3 (Nonlinear Dynamics, Waters, Milford, MA, USA) for peak detection and alignment. A multivariate statistical analysis was performed using the ropls R package from Bioconductor on the Majorbio Cloud Platform. Principal component analysis (PCA) was an unsupervised multivariate statistical analysis method that was used to obtain the whole metabolic data, the overall differences between samples in each group and the degree of variability between samples within the group. Further multivariate analysis using a partial least squares discriminant analysis (PLS-DA) model was performed to determine metabolic changes between comparable groups in positive (+) and negative (−) ion mode. Identified differential metabolites were annotated and mapped into their biochemical pathways in the KEGG compound database (http://www.kegg.jp/kegg/compound/ accessed on 3 April 2023). Heatmap analysis was applied to visualize the annotated differential metabolites in each sample. Metabolic pathways were constructed based on pathway analysis of the differentially expressed metabolites detected in both positive and negative ion modes using MetaboAnalyst 3.0. The *p* value < 0.05 represented significant enrichment of metabolites in a pathway. Venn diagrams were constructed to show the differential metabolites between groups.

### 2.3. Soluble Sugar Analysis

The total soluble sugar content (mg/g) was determined according to the method of Anthrone colorimetry [17]. The sucrose content (mg/g) was determined using gas chromatography tandem mass spectrometry (GC-MS), according to our previous study [18]. A total of 20 mg of tea sample was mixed with 500 μL methanol: isopropanol: water (3:3:2 *V/V/V*) solution, vortexed for 3 min, treated with ultrasound for 30 min, and centrifuged at 14,000 rpm for 3 min at 4 °C. A gas chromatograph (Agilent 8890, Agilent Technologies, Santa Clara, CA, USA) coupled to a mass spectrometer (Agilent 5977B, Agilent Technologies, Santa Clara, CA, USA) with a DB-5MS column (30 m length × 0.25 mm i.d. × 0.25 μm film thickness, J&W Scientific, Folsom, CA, USA) was employed for GC-MS analysis. The identification of sucrose was based on the peak area and standard curve of the samples.

### 2.4. Catechin and Caffeine Analysis

The content and composition of catechins and caffeine (mg/g) were analyzed using high-performance liquid chromatography (HPLC) (Agilent 1200, Agilent Technologies, Santa Clara, CA, USA) [18]. Prepared tea infusions were analyzed using an Agilent Eclipse XD8 C18 column (250 mm × 4.6 mm i.d., 5 μm; Thermo Electron Corporation, Waltham, MA, USA). The injected sample volume was 5 μL with a flow rate of 0.9 mL/min. The UV detection wavelength was set at 278 nm. The identification of catechins and caffeine was based on the peak area and standard curve of the samples.

### 2.5. Amino Acid Analysis

The amino acid and theanine content (mg/g) was analyzed using an HPLC system equipped with a YMC ODS (C18) reverse phase column (YMC-Pack ODS-AQ/S-5 μm/12 nm, YMC CO., LTD. Kyoto Japan) [18]. The mobile phase A was Sodium Acetate (NaAc) (4 mM, pH = 5.5), and the mobile phase B was 80% Acetonitrile solution. The UV detection wavelength was set at 360 nm. The identification of each amino acid was based on the peak area and standard curve of the samples.

### 2.6. Volatile Component Analysis by GC/MS

The volatile components were analyzed using headspace solid-phase microextraction/gas chromatography-mass spectrometry (GC-MS-QP 2010, Shimadzu, Tokyo, Japan), which was conducted according to the method of Liao et al. [19]. A 50/30 μm polydimethylsiloxane/divinylbenzene/carboxen (PDMS/DVB/CAR) SPME fiber (Supelco, Bellefonte, PA, USA) was used for the gas chromatograph, which was preconditioned for 30 min in the injection port at 250 °C before analysis. 1 g of each tea sample was added with 5 mL boiling water, then extracted at 60 °C for 1 h. After the extraction, the extraction head was taken out and inserted into the inlet of the gas chromatograph. GC conditions: DB-5MS elastic quartz capillary column (0.25 μm × 0.25 mm ×30 m); splitless injection mode; heating program: 40 °C for 2 min, raised to 85 °C at 5 °C/min and held for 2 min, increased to 110 °C at 2 °C/min, then increased to 160 °C at 4 °C/min for 1 min, finally increased at 10 °C/min to 230 °C and held for 5 min; the carrier gas was helium (>99.99%) at a constant flow velocity of 1.0 mL/min. MS conditions: electron ionization source; the electron impact mode was generated at 70 eV; temperature for ion source was 230 °C; interface temperature was 230 °C; total ion currents in the 40–400 mass range were recorded to form the chromatograms. The volatiles identification was based on comparing retention indices (RIs) with the published data in the MS library of NIST08 and the internal standard semi-quantitative method (ethyl caprate). A heatmap was used to depict the metabolic changes between two foliar diseases and healthy tea leaves.

### 2.7. Statistical Analysis

The statistical analyses were conducted using IBM SPSS Statistics 20.0 (SPSS Company, Chicago, IL, USA). Three replicates were used for data analysis, which were expressed as mean value ± standard deviation (SD). Significant differences between means were determined by one-way analysis of variance (ANOVA). *p* ≤ 0.05 indicates a significant difference between samples.

## 3. Results and Discussion

### 3.1. Non-Volatile Metabolite Profile of Tea Affected by Diseases

During the process of tea plucking in the field, tender tea shoots infected with blister blight and small leaf spot diseases have usually been harvested together with healthy fresh leaves as raw materials for tea processing; thus, the tea quality is severely affected. Some studies showed that the tea made from blister blight-infected tea leaves had an obvious bitter taste [16], which was inconsistent with our previous study showing that tea leaves infected with blister blight disease had an enhanced sweet taste with a light fishy taste (data not published). The quality of tea is related to a combination of various metabolites [2]. To investigate the differential metabolites between healthy tea leaves (HT) and diseased tea leaves infected with blister blight disease (BB) and small leaf spots (SS) in this study, a nontargeted metabolomic analysis using a UHPLC-Q-TOF/MS system was applied. A total of 4796 peaks were identified in positive mode, while 6146 peaks were identified in negative mode. To obtain an overview of the differences in the metabolites of healthy and diseased tea leaves, unsupervised principal component analysis (PCA) was performed. In the model of the positive (or negative) mode of healthy and diseased tea samples, quality control (QC) samples were closely clustered together and significantly distinguished from the other three groups, which confirmed the reliability of the analysis. As shown in Figure 1A, the trend difference among the group HT, SS and BB was great, and the variation within the group was very small (PC1 = 46.80% and PC2 = 24.00%) (Figure 1A). The PCA score plot not only showed obvious different metabolic differences between healthy and diseased tea leaves but also showed significant differences between BB and SS, indicating that both blister blight and small leaf spot diseases had a greatly different influence on the metabolites of tea. Another multivariate analysis technique—partial least squares discriminant analysis (PLS-DA)—was also applied to reveal significant differences among the three groups (Figure 1B). Both PLS-DA and PCA models in this study showed good reliability and repeatability. A heatmap was used to visualize the relative variation of different components in HT, SS and BB, in which each column represented a tea sample, and each row represented a specific metabolite. A total of 519 differential metabolites were detected among the three groups in positive (+) and negative (−) ion modes, as shown in Appendix A. Color coding was graded from red to purple with the relative intensity shift from high to low, respectively. A red box indicates that a metabolite was at a higher level than the mean level in a sample, and a purple box means the metabolite was at a lower level (Appendix A).

### 3.2. Differential Non-Volatile Metabolite Analysis

According to the screening results of differential metabolites from HT, SS and BB, Venn diagrams in positive (+) ion mode were constructed (Figure 2A,B). As shown in Figure 2A, the round dots connected by straight lines in the diagram represent the metabolites common to multiple metabolic sets, the round dots without straight lines represent the metabolites unique to the metabolic set, and the numbers shown above the column of the bar chart are the corresponding number of metabolites. Of the 472 differential metabolites, 450 metabolites were common across the three groups HT, SS and BB, 3 were common between HT and SS, and 7 were common between BB and SS; only 6 differential metabolites were exclusively found in BB and the same number of differential metabolites in SS. Compared with HT, the same number (13) of differential metabolites was exclusively identified in BB or SS (Figure 2A). In negative (-) ion mode, a total of 482 differential metabolites were detected in all groups, 468 were the common differential metabolites, and only 2 were exclusively found in SS (Appendix A). To further determine the metabolite changes induced by blister blight and small spot diseases, a Venn plot was constructed based on the differential metabolites from SS_vs_HT, BB_vs_HT and BB_vs_SS (Figure 2B). A total of 315 and 335 differential metabolites were detected in SS_vs_HT and BB_vs_HT, respectively. Furthermore, 95 differential metabolites (accounting for 18.3%) were common among the three groups, and 53 (10.2%), 35 (6.74%) and 57 (11.0%) differential metabolites were exclusively found in SS_vs_HT, BB_vs_HT and BB_vs_SS, respectively (Figure 2B). Among the metabolites, compared with healthy tea shoots, 205 and 197 metabolites were increased in SS and BB, respectively (Figure 2C), revealing that most of the measured metabolites were increased after the disease infection. On comparing the differential metabolites between BB and SS, a total of 338 differential metabolites were found in BB_vs_SS, and of those, 145 metabolites were increased, while 193 were decreased (Figure 2C).

### 3.3. Classification of Annotated Differential Metabolites

In response to pathogen infection or insect attack, tea flavor and quality are significantly influenced by a change in the quality-related metabolites [5,6,7,10,20]. The quality of tea can be determined by its chemical composition. All the annotated differential compounds detected between healthy and diseased tea leaves were classified into eight categories, including flavonoids, phenylpropanoids, alkaloids, amino acid-related compounds, terpenoids, polyketides, shikimate/acetate-malonate pathway-derived compounds, fatty acid-related compounds and others in the KEGG compound database (Figure 3A). The classification of KEGG compounds was based on the biological functional hierarchy of metabolites. Among the metabolites in the second category, flavonoids and monolignols were significantly enriched and changed (Figure 3A). Through metabolic pathway mapping, the significantly changed metabolites in response to disease infection in the BB and SS groups were mainly involved in metabolism pathways, including phenylpropanoid biosynthesis, ABC transporters, glutathione metabolism and arginine and proline metabolism (Figure 3B). Enrichment rate is the ratio of the metabolite number enriched in the pathway to the metabolite number annotated in the pathway. However, based on the metabolomics data and KEGG mapping, differential metabolites, especially down-regulated metabolites, were mainly involved in tea quality and host defense-related biosynthesis metabolism involving phenylpropanoid and flavonoid biosynthesis pathways (*p* < 0.05) (Figure 3B). These results suggest that the metabolomics analysis enables good characterization of the differences in metabolism pathways between healthy and infected tea.

The enriched monolignols are usually synthesized from phenylalanine via the general phenylpropanoid and monolignol-specific pathways [21]. Sinapyl alcohol and hydroxycinnamyl alcohols (or monolignols) are the main building blocks of lignin, and the biosynthesis of lignin can be induced by various biotic and abiotic stresses, such as pathogen infection, wounding, etc. [21]. In this study, 11 compounds (monolignols) involved in phenylpropanoid biosynthesis were detected, and six main monolignols, including sinapyl alcohol, 4-hydroxy-3-methoxycinnamaldehyde, trans-o-coumaric acid 2-glucoside, ferulic acid, trans-cinnamic acid and 4-hydroxycinnamic acid, were all significantly increased in the BB and SS groups compared with HT (Figure 3C), suggesting that the major monolignols were remarkably induced in response to both diseases. Sinapyl alcohol was the end product in the phenylpropanoid biosynthesis pathway (Figure 3D). Chlorogenic acid was the product of caffeic acid by the metabolic intermediate caffeoyl-CoA, ferulic acid also could be converted from caffeic acid, and 5-hydroxyferulate was the metabolic product of ferulic acid (Figure 3D). The contents of chlorogenic acid and caffeic acid were significantly increased in BB but decreased in SS (Figure 3C), suggesting that the metabolites play different roles in tea plants in response to pathogen infection.

### 3.4. Changes in the Main Taste-Related Metabolites in Diseased Tea Leaves

Flavonoids, a class of plant polyphenols derived from plant secondary metabolism with defense functions, are widely found in plants. Some characteristic flavonoid compounds, such as catechins, flavone glycosides, proanthocyanidin dimers, flavonol glycosides and hydrolyzable tannins in tea leaves, can be significantly enhanced by infestation with the green leafhopper [22]. Catechins are the main flavonoids in tea, which are synthesized from the flavonoid biosynthesis pathway and phenylpropanoid pathway. Tea geometrids and green leafhoppers are the two major pests occurring on tea plants worldwide, and the effect of the two insects on plant hormones, tea quality-related volatiles and non-volatiles have been widely studied [22,23]. However, few studies have centered on the metabolite changes in tea plants due to pathogen infection. *Colletotrichum*, which causes brown blight disease, is the most common pathogen occurring in tea plants, severely affecting the growth of the tea plant [24,25]. Wang et al. [10] found that the levels of (−)-epigallocatechin gallate (EGCG), (+)-catechin (C) and caffeine were induced in *C. fructicola*-resistant leaf tissues. Catechins, the primary bitter and astringent substances in tea [22], are the main flavonoids in tea and are classified as esterified or non-esterified catechins. Four esterified and four non-esterified catechins were detected in our study by HPLC, and the concentration of esterified catechins was much greater than that of non-esterified catechins. As shown in Figure 4A, compared with HT, total catechins in BB and SS were decreased by 23.0% and 6.00%, respectively. The main esterified type, EGCG, and (−)-epicatechin gallate (ECG) accounted for 72.0%, 58.0% and 57.0% of the total catechins in HT, BB and SS, respectively, and the levels of EGCG (Figure 4B) and ECG (Figure 4C) were all significantly decreased in BB and SS compared with those in HT. In diseased tea samples, the content of EGCG and ECG in SS (67.1 and 18.3 mg/g) was significantly higher than that in BB (58.1 and 9.04 mg/g) (Figure 4B,C). EGCG and ECG were the abundant catechins in tea and could be converted from (−)-epigallocatechin (EGC) and (−)-epicatechin (EC), respectively, via the sequential action of flavan-3-ol gallate synthase [26]. However, compared with HT, the contents of EGC in SS and BB were remarkably increased by 49.0% and 39.0%, respectively, and EC had the highest content in SS while the lowest content in BB (Figure 4D,E), indicating that both EGC and EC were significantly induced in small leaf spot disease-infected tea and only EGC was induced by blister blight disease. Various constitutive and induced plant phenolic compounds have been proven to contribute to defense against microbial pathogens [20]. However, in our study, the amount of total catechins and most individual catechins were significantly reduced after infection with both blister blight and small leaf spot diseases. Similar results reported by previous studies showed that the incidence of blister blight disease resulted in a decline in the polyphenol content and enzymes in tea leaves, which adversely affected various quality parameters [16,27].

Taste is one of the most important quality characteristics of tea, which is closely related to the characteristic chemicals, including tea polyphenols, catechins, caffeine, soluble sugar and L-theanine [3]. It has been revealed that catechins and caffeine mainly contribute to bitterness and astringency [28], while amino acids and soluble sugar contribute to the umami and sweet taste, respectively, which are responsible for the unique flavors of tea [29]. As another bitter-taste substance, the caffeine content in both diseased tea leaves was much lower than that in HT, but it was higher in BB than in SS (Figure 4F), indicating that caffeine is not positively associated with infection by blister blight and small leaf spot diseases. Previous studies suggested that caffeine can be induced by *C. fructicola* infection [10], which may exert its antimicrobial effects by damaging the cell wall and membrane of pathogens. These results suggested that the metabolite changes in tea are differentially affected by various types of diseases. Soluble sugar is an important flavor substance in tea. The content of soluble sugar in HT was about 28.3 mg/g, and the accumulation was significantly increased in SS and BB by 24.4% and 77.3%, respectively (Figure 4G). Sugars may play a vital role in plant defenses against pathogens and can be induced by pathogen infection [30,31], which was in accordance with our results. Although the accumulation of soluble sugar in this study was significantly enhanced by infection with *D. segeticola* and *E. vexans*, the level in BB was much higher (increased by 43.0%) than that in SS (Figure 4G), suggesting that soluble sugar—especially sucrose—is remarkably induced by tea blister blight disease, which is also positively associated with the enhanced sweet taste. Sucrose—belonging to disaccharides—is an abundant sugar in tea. Compared with HT (6.20 mg/g), the content of sucrose in SS (11.0 mg/g) and BB (22.1 mg/g) was remarkably enhanced, and the content in BB was much higher than that in SS (Figure 4H).

Different kinds of amino acids may have different flavors, and theanine, as the main component of amino acids, endows tea infusions with umami taste [4,19,23]. The changes in amino acids and theanine in two diseased tea leaves showed similar trends (Figure 4I,J). Theanine, as the most abundant non-protein amino acid in tea, accounted for 74.0%, 63.0% and 70.0% of the total amino acids in HT, SS and BB, respectively. Compared with HT, the content of free amino acids in SS and BB was sharply decreased by 43.0% and 20.0%, respectively, and the theanine content of SS and BB was significantly decreased by 51.0% and 22.0% (Figure 4I,J), respectively, indicating that amino acids and theanine are both negatively affected by blister blight and small leaf spot diseases but are less affected by blister blight disease. The phenol-ammonia ratio is the ratio of tea polyphenols to free amino acids, which is closely related to the fresh taste of tea. The phenol-ammonia ratio in SS was much higher than that in BB and HT (Figure 4K).

In general, the analysis of the non-volatile metabolites showed that the accumulation of catechins, caffeine, soluble sugar and amino acids greatly contributed to the quality and unique flavor of tea [1]. In our study, the contents of sweet and umami-related soluble sugar, free amino acids and theanine in BB were much higher than those in SS, while total catechins, EGCG, ECG, EGC and EC were the opposite. However, the levels of most metabolites, including amino acids, total catechins and caffeine, were still lower than those in healthy tea leaves (Figure 4). Therefore, the quality of tea was significantly affected by infection with blister blight and small leaf spot diseases by changing the content and composition of metabolites in tea.

### 3.5. Volatile Profiles in Diseased Tea Leaves

Volatile compounds only account for 0.01% of dry weight in tea, but they play a vital role in tea quality, which is related to the content and composition of volatile metabolites [5]. In recent years, the interaction between tea plants and insects has largely focused on volatile compounds [5,32]. Tea plants usually release a large amount of volatile metabolites, such as terpenoids, green leaf volatiles and other aromatic compounds, in response to attack by invaders such as pathogens and pests [5,32,33,34]. These induced volatile metabolites may have the function of plant protection and are also important tea-quality components [7]. Tea plants mildly infested with tea green leafhoppers can make tea more aromatic, and tea made from this kind of tea leaves has a characteristic flavor, forming a famous oolong tea, “Oriental Beauty” [5]. However, so far, there have been limited reports on the volatiles profile of tea infected with disease. In this study, a comprehensive analysis was conducted by GC-MS to reveal the content and composition of volatile compounds in tea leaves infected with blister blight and small leaf spot diseases. As shown in Figure 5, 64 volatiles were detected in healthy and diseased tea samples. Red and purple boxes represent the metabolites at higher and lower levels than the mean level in each sample. Among them, 42, 39 and 52 volatile compounds were found in HT, SS and BB, respectively (Figure 5A), and the corresponding total amounts of volatiles were 79.3, 29.3 and 46.3 μg/g, respectively (Figure 5B). It was obvious that although the number of volatile metabolites in BB was higher than that in HT, the content of metabolites in both SS and BB was significantly decreased by 63.0% and 42.0%, respectively, compared with HT. Among the volatiles, 28 were commonly detected volatiles in HT, SS and BB, 7 were newly detected metabolites in SS but at very low amounts compared with HT, and up to 20 new metabolites were detected in BB. The content of each compound was all less than 1.00 μg/g, suggesting that a large number of volatiles were induced after pathogen infection but at low levels (Figure 5C). This study was the first to systematically illustrate the volatiles profile of tea affected by blister blight and small leaf spot diseases, and indicates that the number and content of volatile compounds in tea leaves were significantly changed by infection with *E. vexans* and *D. segeticola*. Some of the results of this study were not consistent with a previous study on blister blight disease by Zhang et al. [32], in which a total of 56 volatile compounds were identified on *E. vexans*-infected tea plants (*Camellia sinensis* var. *assamica*) by using SDE/GC/MS, and the amount of volatile compounds in infected tea leaves was higher than that in healthy tea leaves. The different results may be due to the different tea varieties and growth environment. Different tea cultivars, especially large-leaf and small-leaf tea varieties, usually have distinctly different volatile characteristics, which are also influenced by the growth environment.

### 3.6. Variation of Volatiles in Diseased Tea Leaves

In tea leaves infected with small leaf spots, all the volatile compounds belonged to six chemical groups that consisted of 15 alcohols (accounting for the total content in SS, 38.7%), 8 aldehydes (22.2%), 7 aromatic hydrocarbons (13.2%), 6 esters (16.6%), 2 ketones (7.10%), and 1 nitrogen compound (2.22%) (Figure 6A). Although the number of volatile compounds in HT and SS was similar, the amount of total and individual compounds was obviously decreased (Figure 6A). In BB, 7 chemical groups were found, including 15 alcohols (17.7%), 13 aldehydes (41.6%), 7 aromatic hydrocarbons (15.9%), 9 esters (18.1%), 3 ketones (2.42%), 4 nitrogen compounds (3.97%) and 1 acid (0.35%) (Figure 6A). In HT, esters (30.4%), alcohols (22.8%) and aldehydes (15.9%) were the major volatiles, accounting for 69.1% of the total amount of volatiles, while alcohols and aldehydes were the most abundant in SS and BB, respectively. The results indicate that the types and amounts of volatile substances were highly influenced by infection with small leaf spot and blister blight diseases.

In HT, (Z)-3-hexenyl acetate (11.3 μg/g), (Z)-3-hexenyl hexanoate (10.9 μg/g), nonanal (8.53 μg/g), linalool (6.96 μg/g) and heptanal (4.38 μg/g) were the major compounds in this study (Figure 6B). In contrast, the total amount of the volatiles in SS was only 29.3 μg/g, and only 10 compounds had a content more than 1.00 μg/g. Among these compounds, linalool (3.82 μg/g) and nonanal (2.90 μg/g) were the most abundant compounds in SS (Figure 6B). However, although some chemicals such as linalool and nonanal still were the major compounds in SS, the contents were significantly decreased compared with those in HT, suggesting that tea quality may be significantly affected by the concentration decrease and composition change of volatile compounds. So far, there have been no reports about the quality-related metabolites of tea affected by small leaf spot disease. In BB, nonanal (8.23 μg/g), styrene (5.09 μg/g), heptanal (3.51 μg/g), formic acid phenylmethyl ester (3.47 μg/g) and (Z)-3-hexenyl acetate (3.22 μg/g) were the most abundant volatile compounds (Figure 6B). The contents of nonanal, heptanal and formic acid phenylmethyl ester in BB were similar to those in HT, while the content of styrene was highly increased by 194% and (Z)-3-hexenyl acetate was significantly decreased from 11.3 to 3.22 μg/g (Figure 6B). These results indicate that styrene could be significantly induced by infection with blister blight disease, while (Z)-3-hexenyl acetate was negatively affected, which was not consistent with previous studies that showed terpenes, azotic, sulfur, aromatic and green leaf volatiles were the main volatiles in blister blight-infected tea leaves [17,32]. A previous study showed that (Z)-3-hexenyl hexanoate and (Z)-3-hexenyl acetate can be elicited in tea plants attacked by tea geometrid (*Ectropis obliqua*) [34], and styrene is involved in the resistance to tea green leafhopper (*Empoasca flavescens*) [33]. Some other volatiles, such as (E)-nerolidol, were detected in much lower amounts in our study (Figure 5), and were reported to be a volatile signal that participated in the defense of tea plants against both a herbivore (*Empoasca*) and a pathogen (*C. fructicola*) [35]. Therefore, the concentration and composition of volatile substances were significantly changed by both small leaf spot and blister blight diseases (Figure 7), finally maybe resulting in a change in tea quality. Further studies on the volatile signals in response to blister blight and small leaf spot diseases will be important for disease control and tea quality enhancement as a fungicide or inducer.

## 4. Conclusions

Blister blight and small leaf spot diseases are important alpine diseases that mainly attack tender tea leaves, affecting tea quality, and they occur widely throughout alpine tea gardens worldwide, especially in southwest China, including Guizhou, Sichuan province and Chongqing city in recent years (Figure 7). In our study, the effect of blister blight and small leaf spot diseases on the non-volatiles and volatiles in tea were systematically investigated and compared. Among the non-volatile metabolites, flavonoids and monolignols were significantly enriched and changed. Six main monolignols involved in phenylpropanoid biosynthesis were significantly induced in infected tea leaves. The level of sweet- and umami-related soluble sugar, sucrose, free amino acids and theanine were much higher in BB, while bitter and astringent taste-related catechins and derivatives were much higher in SS, indicating tea flavors are differentially influenced by these two diseases. The accumulation of total volatile compounds and most of the individual volatile compounds in tea were significantly decreased by infection with both diseases; however, styrene was significantly induced in the blister blight-infected tea leaves. In summary, the occurrence of two diseases significantly changed the composition and content of non-volatile and volatile metabolites of tea.

## Figures and Tables

**Figure 1 foods-12-01568-f001:**
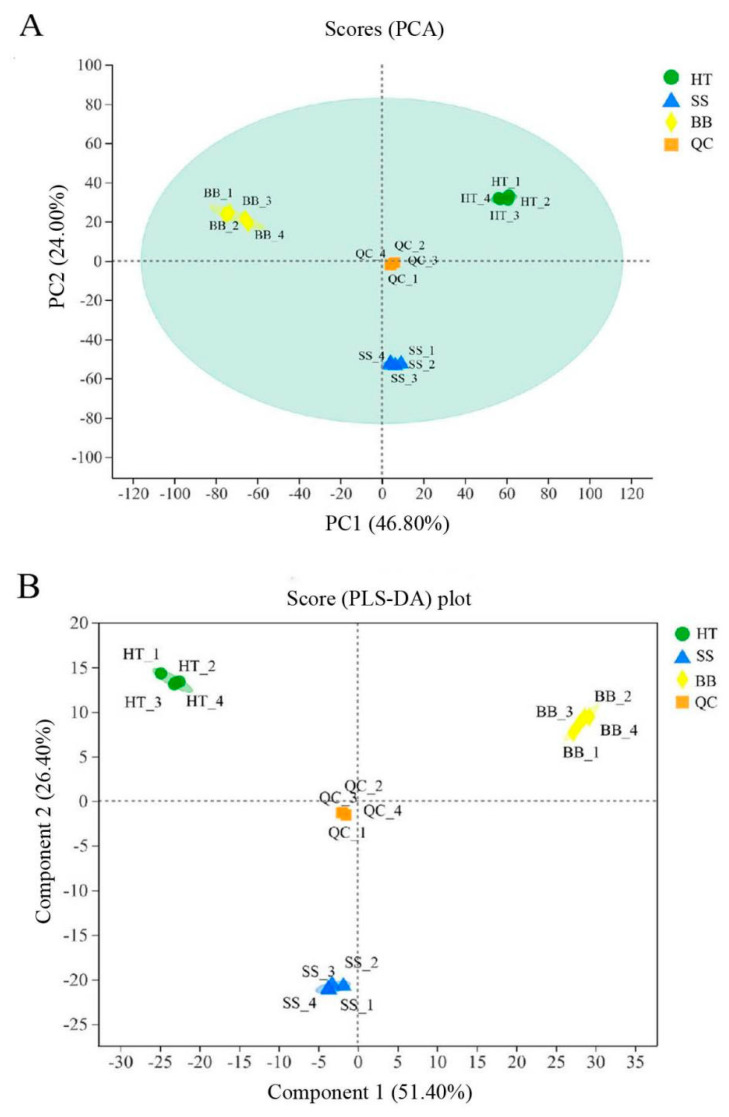
The PCA score plot (**A**) and PLS–DA score plot (**B**) analysis of the healthy and diseased tea samples. HT: Healthy tea shoots, SS: Tea shoots infected with small leaf spot disease, BB: Tea shoots infected with blister blight disease.

**Figure 2 foods-12-01568-f002:**
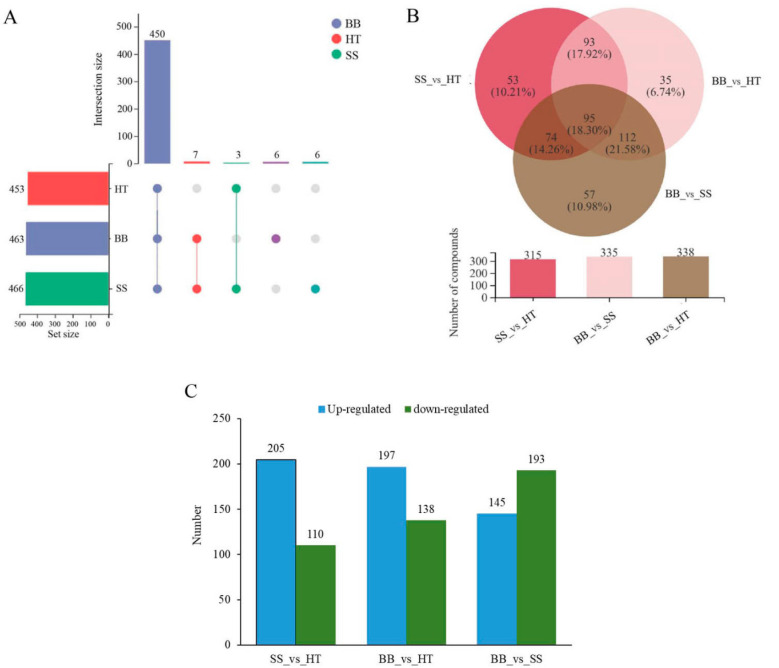
Overview of the differential components in healthy and diseased tea samples. (**A**,**B**) Overlapping relationship of differential metabolites among different comparable samples shown in Venn diagrams. (**C**) Numbers of the down-regulated and up-regulated metabolites in different tea samples. HT: Healthy tea shoots, SS: Tea shoots infected with small leaf spot disease, BB: Tea shoots infected with blister blight disease.

**Figure 3 foods-12-01568-f003:**
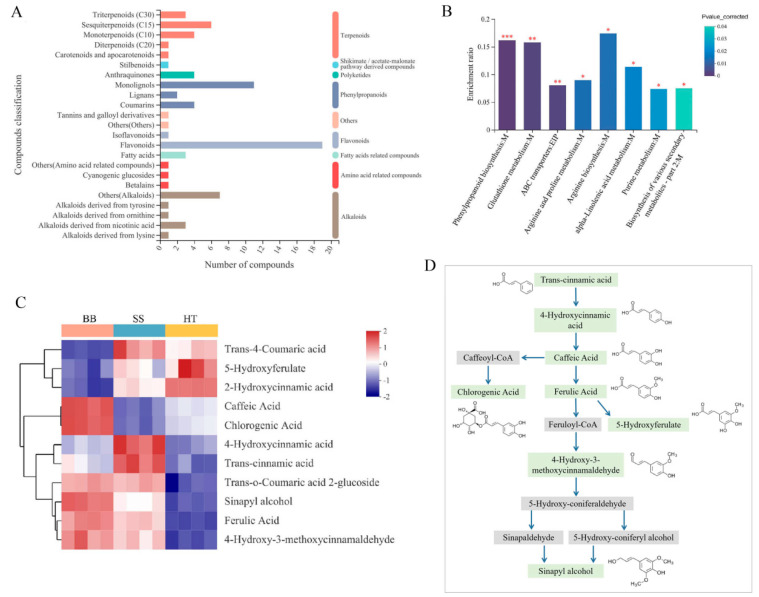
Metabolite analysis of the differential metabolites of tea in healthy and diseased tea samples. (**A**) Classification of differential metabolites among HT, BB and SS in the KEGG compound database. (**B**) KEGG enrichment analysis of the metabolic pathway in diseased tea. (**C**) The changes in the enriched monolignols in tea. *p* < 0.05 is showed as “*”, *p* < 0.01 is showed as “**” and *p* < 0.001 is showed as “***”. (**D**) The enriched monolignols involved in phenylpropanoid biosynthesis metabolism. The metabolites marked in light green were detected in our study.

**Figure 4 foods-12-01568-f004:**
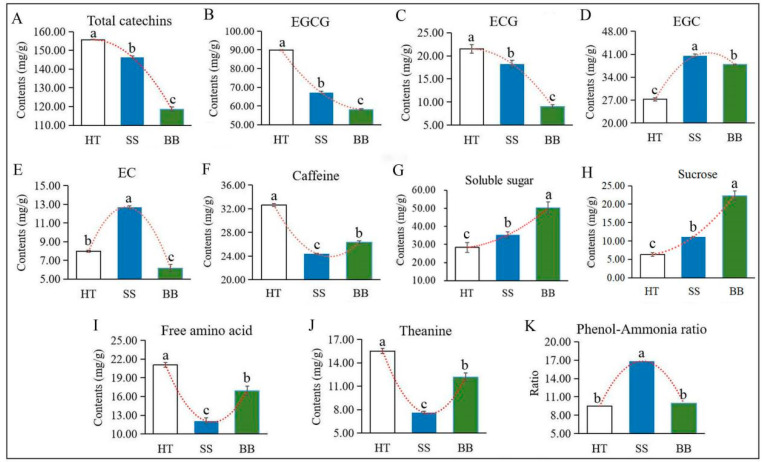
The contents of main quality-related metabolites in HT, BB and SS. (**A**–**K**) The content of total catechins (**A**), EGCG ((−)-epigallocatechin gallate) (**B**), ECG ((−)-epicatechin gallate) (**C**), EGC ((−)-epigallocatechin) (**D**), EC ((−)-epicatechin) (**E**), caffeine (**F**), soluble sugar (**G**), sucrose (**H**), free amino acids (**I**), theanine (**J**) and the phenol-ammonia ratio (**K**). The different letters above the column indicate significant difference (*p* ≤ 0.05). HT: Healthy tea shoots, SS: Tea shoots infected with small leaf spot disease, BB: Tea shoots infected with blister blight disease.

**Figure 5 foods-12-01568-f005:**
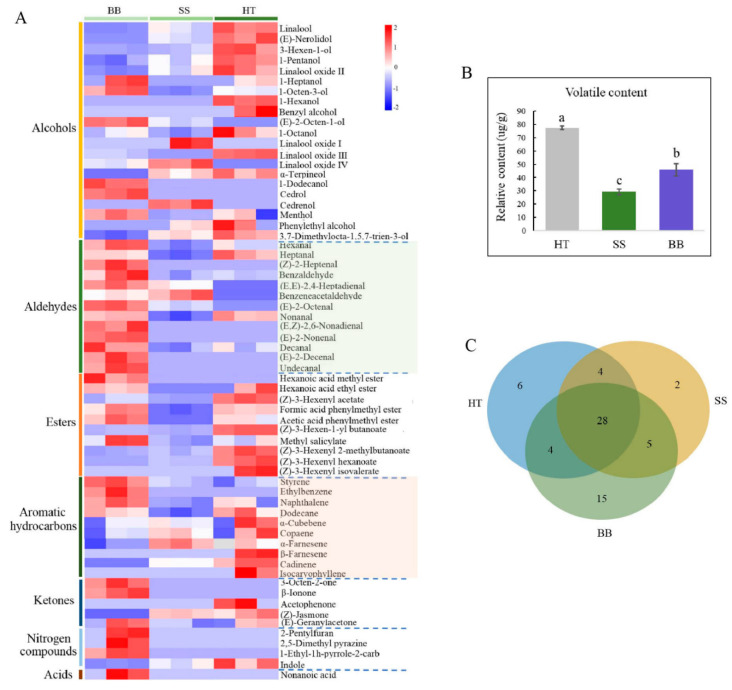
Volatiles profile (**A**), content (**B**) and overlapping relationship of differential volatile metabolites (**C**) in tea affected by blister blight and small leaf spot diseases. HT: Healthy tea shoots, SS: Tea shoots infected with small leaf spot disease, BB: Tea shoots infected with blister blight disease.

**Figure 6 foods-12-01568-f006:**
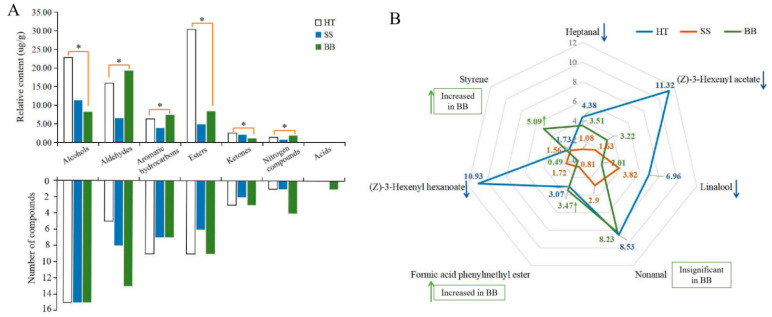
The number and proportion of the volatile category in healthy and diseased tea samples (**A**) and the main volatile compounds in healthy and diseased tea (**B**). *p* ≤ 0.05 is shown as “*”. HT: Healthy tea shoots, SS: Tea shoots infected with small leaf spot disease, BB: Tea shoots infected with blister blight disease.

**Figure 7 foods-12-01568-f007:**
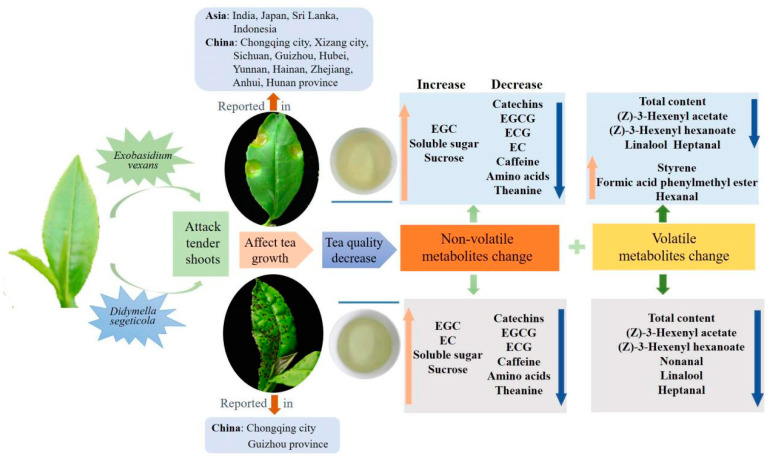
The overview of tea influenced by blister blight and small leaf spot diseases.

## Data Availability

The data presented in this study are available in the article and Appendix A.

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
