# Peer review of "Metabolomics Analysis Reveals the Effect of Two Alpine Foliar Diseases on the Non-Volatile and Volatile Metabolites of Tea"

_foods, 2023, doi:10.3390/foods12081568_

Round 1

Reviewer 1 Report

Material and methods

-  No information was given in what units the content of the analyzed compounds was expressed (total soluble sugar, sucrose, catechins, caffeine, amino acids, theanine, volatile components)

Line 186, 189-190 – Fig. S1 not included in the manuscript

Line 208 – Fig. S2 not included in the manuscript

References

According to the Instructions for Authors:

- abbreviated names of journals should be given

- the year should be written in bold font

- volume number should be written in italics

Line 478 - The Latin name should be written in italics

Line 487 – Author “Dong” is missing

Line 488 – “onukii” should be written in italics

Line 496 – Author “Yang” is missing

Line 498 – Author “Ynag” - check

Line 534 - The Latin name should be written in italics

Line 544 – please check the volume number

Line 545-546 – please check the journal name

Line 548 – “…Sinensis …” – replace “S” with “s”

Line 553-554 – check authors’ names, Authors: Zhang, Li, Chen and Xin are missing

Author Response

1.Comments: Material and methods

No information was given in what units the content of the analyzed compounds was expressed (total soluble sugar, sucrose, catechins, caffeine, amino acids, theanine, volatile components)

-Line 186, 189-190 – Fig. S1 not included in the manuscript. 

-Line 208 – Fig. S2 not included in the manuscript

Response: Thanks to the Reviewers’ thoughtful and good comments. The content units of the analyzed compounds were shown in Fig. 4, and we also added the units of the contents of compounds in the section Material and methods. We are sorry for our careless and Fig S1 and Fig. S2 had been added an the end of the manuscript.

2.Comments: References

- abbreviated names of journals should be given

- the year should be written in bold font

- volume number should be written in italics

- Line 478 - The Latin name should be written in italics

- Line 487 – Author “Dong” is missing

- Line 488 – “onukii” should be written in italics

- Line 496 – Author “Yang” is missing(没有找到)

- Line 498 – Author “Ynag” - check

- Line 534 - The Latin name should be written in italics

- Line 544 – please check the volume number

- Line 545-546 – please check the journal name

- Line 548 – “…Sinensis …” – replace “S” with “s”

- Line 553-554 – check authors’ names, Authors: Zhang, Li, Chen and Xin are missing

Response: Thanks to the Reviewers’ detailed comments. We are sorry for so many mistakes in references of the manuscript, we have revised the References according to the Reviewers’ suggestions and highlighted by using red-colored text.

Reviewer 2 Report

This manuscript describes the effects of two alpine foliar diseases on the non-volatile and volatile metabolites of tea through metabolomics analysis. This manuscript is written well with important information for the scientific public. I have following suggestions to improve this.

Line 20-23: The sentence is too long. Authors may revise or break that into 2 sentences for clarity.

Line 29-33. Run on sentence needs to be re-written.

Line 38-41. Re-write for clarity.

Line 58-64. Long sentence and may re-written for clarity.

Line 226-268: More supporting literature could be added.

Line 397-432: More supporting literature could be added/cited.

Overall, the content of the manuscript is good and English grammar editing is recommended.

Author Response

1.Comments:

- Line 20-23: The sentence is too long. Authors may revise or break that into 2 sentences for clarity.

Response: Thanks to the Reviewers’ comments, the sentence in Line 20-23 has been revised as follows: Volatiles analysis showed that volatiles content in SS and BB were all significantly decreased, and styrene was significantly induced in blister blight infected tea leaves. The results indicated that the type and amount of volatiles were highly and differentially influenced by the infection of the two alpine diseases.”

2.Comments:

- Line 29-33. Run on sentence needs to be re-written.

Response: Thanks to the Reviewers’ comments, the sentence has been re-written as follows: Numerous studies have validated that biotic and abiotic stresses including pathogen infection (such as Colletotrichum, Exobasidium), insect attack (tea geometrid, tea green leafhoppers), mechanical damage and environment conditions can significantly affect the quality of tea by changing the content and composition of metabolites, especially the volatile compounds in tea plants [4,5,6,7].

3.Comments:

- Line 38-41. Re-write for clarity.

Response: Thanks to the Reviewers’ comments, and this sentence was re-written as follows: Tea plants have abundant caffeine and phenolic compounds which act as a vital role in disease resistance, and the amount of these compounds in tea plants were usually increased in response to pathogen infection [9,10].

4.Comments:

- Line 58-64. Long sentence and may re-written for clarity

Response: Thanks to the Reviewers’ comments, and this sentence was re-written as follows: To reveal and compare the effect of blister blight and small leaf spot diseases on the non-volatile and volatile metabolites of tea, the differential metabolites between healthy and diseased tea leaves were comprehensively investigated. The ultra-high performance liquid chromatography-quadrupole-time of flight mass spectrometry (UHPLC-Q-TOF/MS), high performance liquid chromatography (HPLC) and gas chromatography tandem mass spectrometry (GC-MS) were applied for nontargeted and targeted metabolomic analysis.   

5.Comments:

- Line 226-268: More supporting literature could be added.

Response: Thanks to the Reviewers’ comments, and more supporting literature have been added follows: In response to pathogen infection or insects attack, tea flavor and quality were significantly influenced by the change of the quality related metabolites [5-7,10,20].

6.Comments:

- Line 397-432: More supporting literature could be added/cited.

Response: Thanks to the Reviewers’ comments. The contents in Line 397-432 all are our results, and so far, there were no reports about the quality related metabolites of tea affected by small leaf spot disease. Therefore, no literature were cited in this section, however, there are some literature after the results in our study.

Reviewer 3 Report

Dear Authors,

The manuscript (foods-2286523) submitted for review is interesting and valuable. I recommend it to correct the figures.

Authors, Please note and address the following comments:

The introduction, results, discussion and conclusion are well written.

The research methodology is described in detail and there is a possibility of replicating of these studies by other authors.

Interesting study, but based on only 3 replicates of tea samples or 3 replicates of each analysis of healthy tea leaves and leaves with blister blight and small leaf spots diseases. It should be mentioned in limitations.

Technical notes:

Only two of the seven Figures are legible. Figures 2, 3, 5, 6 should be corrected as they are illegible.

It seems that the abbreviations HT, SS, BB should also be explained under figure 5, they are only explained under figure 1.

HT: Healthy tea shoots, SS: Tea shoots infected with small leaf spots disease, BB: Tea shoots  infected with blister blight disease.

References: References are not cited according to journal rules. Publications from MDPI provide information on how to properly cite. Authors may also find this information in the authors' guide.

 Despite my comments, I am pleased to recommend this manuscript for minor correction.

Reviewer

Author Response

1.Comments:

Interesting study, but based on only 3 replicates of tea samples or 3 replicates of each analysis of healthy tea leaves and leaves with blister blight and small leaf spots diseases. It should be mentioned in limitations.

Response: Thanks to the Reviewers’ comments. Three replicates were used for data analysis, which were expressed as mean value ± standard deviation (SD). The contents were mentioned in 2.7. Statistical Analysis.

2.Comments: Technical notes

Only two of the seven Figures are legible. Figures 2, 3, 5, 6 should be corrected as they are illegible.

It seems that the abbreviations HT, SS, BB should also be explained under figure 5, they are only explained under figure 1.

HT: Healthy tea shoots, SS: Tea shoots infected with small leaf spots disease, BB: Tea shoots infected with blister blight disease.  

Response: Thanks to the Reviewers’ suggestions. We have improved the quality of Figures 2, 3, 5, 6 and they were the highest quality we can provide. The full name of the abbreviations HT, SS, BB have been explained under figure 5.

  1. Comments: References

References are not cited according to journal rules. Publications from MDPI provide information on how to properly cite. Authors may also find this information in the authors' guide.

Response: Thanks to the Reviewers’ comments. We are so sorry for the mistakes in References, and incorrected format has been amended and examined, both the modification were covered in red.